# Establishing a core outcome set for treatment of uncomplicated appendicitis in children: study protocol for an international Delphi survey

Max Knaapen,[1] Nigel J Hall,[2] Johanna H van der Lee,[3] Nancy J Butcher,[4] Martin Offringa,[5] Ernst W E Van Heurn,[1] Roel Bakx,[1] Ramon R Gorter,[1] On behalf of the Paediatric Appendicitis COS development group

**Correspondence to**
Max Knaapen;
M.Knaapen@amc.uva.nl

## ABSTRACT

**Introduction** Appendicitis is a global disease affecting roughly 1 in every 12 people in the world, with the highest incidence between ages 10 and 19 years. To date, a wide variety of health outcomes have been reported in randomised controlled trials and meta-analyses evaluating treatments for appendicitis. This is especially the case in studies comparing non-operative treatment with operative treatment. A set of standard outcomes, to be reported in all future trials, is needed to allow for adequate comparison and interpretation of clinical trial results and to make data pooling possible. This protocol describes the development of such a global core outcome set (COS) to allow unified reporting of treatment interventions in children with acute uncomplicated appendicitis.

**Methods and analysis** We use current international standard methodology for the development and reporting of this COS. Its development consists of three phases: (1) an update of the most recent systematic review on outcomes reported in uncomplicated paediatric appendicitis research to identify additional outcomes, (2) a three-step global Delphi study to identify a set of core outcomes for which there is consensus between parents and (paediatric) surgeons and (3) an expert meeting to finalise the COS and its definitions. Children and young people will be involved through their parents during phase 2 and will be engaged directly using a customised face-to-face approach.

**Ethics and dissemination** The medical research ethics committee of the Academic Medical Center Amsterdam has approved the study. Each participating country/research group will ascertain ethics board approval. Electronic informed consent will be obtained from all participants. Results will be presented in peer-reviewed academic journals and at (international) conferences.

**Trial registration number** COMET registration: 1119

## INTRODUCTION

Appendicitis is a common gastrointestinal disease affecting roughly 1 in every 12 people in the world, with the highest incidence between ages 10 and 19 years.[1 2] While the incidence varies from country to country,

### Strengths and limitations of this study

► This protocol describes an international online Delhi study that should result in a globally relevant set of core outcomes for paediatric uncomplicated appendicitis.
► The protocol was developed in conjunction with an international steering committee and patient representation, and follows all relevant core outcome set (COS) development guidelines and standards.
► This study involves parents and patients in deciding what to measure in future uncomplicated appendicitis research.
► The involvement of young people in COS development requires a customised approach. This protocol addresses this issue and describes a direct face-to-face involvement.
► Because of the global and multilingual aspect of the study, there will be a limited consensus discussion with only selected individuals. Also, due to feasibility, the direct face-to-face engagement of young people will take place only in selected countries.

appendicitis is a global disease.[3] In the last decade, there have been several developments in the treatment of appendicitis in children, with the most recent being non-operative treatment (NOT) for acute uncomplicated appendicitis. Studies investigating the effectiveness of NOT in children show promising results.[4–7] However, the selected primary (and secondary) outcomes vary widely, as reflected in recent systematic reviews assessing the efficacy and safety of NOT, which may contribute to their contradictory conclusions.[4–8] In the systematic review by Georgiou et al,[4] the need for universal outcome selection and reporting in appendicitis studies is emphasised. In general, it is recognised that clinical trials in children often lack outcomes that are appropriately chosen for this particular population.[9]

Inconsistent selection and reporting of outcomes limit the ability to adequately compare and interpret clinical trial results. Furthermore, it hampers data pooling and subsequent meta-analysis. It also increases the risk of selective outcome reporting, a form of publication bias. This, in turn, jeopardises the validity of results from individual trials, which feeds into subsequent systematic reviews[10] and meta-analyses, which are by nature retrospective and therefore liable to various risks of bias.[11 12]

As demonstrated by Hall *et al* in 2015, a wide variety of outcomes has been reported in randomised controlled trials (RCTs) and meta-analyses reporting on the treatment of appendicitis in children.[13] In the 63 included studies, a total of 115 different outcomes were reported.[13] Hall *et al* proposed the development of a core outcome set (COS), which is a standardised collection of outcomes that should be measured and reported in all future trials.[14] Recently, a study protocol was published for developing such a COS in the UK.[15] Because of the differences between countries in treatment practices, resources and cultural aspects, it was decided, in conjunction with the UK COS research group, that there is a need for an international COS to be used in all trials assessing the treatment of acute uncomplicated appendicitis in children. The development of the current international protocol was performed in conjunction with the UK research group. Its principal investigator (NJ Hall) has been involved in its development and is part of the study management group.

Outcomes considered important by patients and families are essential to a meaningful and complete COS.[16] That is why parents and patients play a central role in the consensus process as a stakeholder group. Parent and patient representation was ensured through involvement of the Dutch patient and parent foundation: 'Children and Hospital'. A representative from this group provided feedback from the perspective of parents and children in several stages of the protocol development. They are also involved in the development of a face-to-face methodology for engaging children in this COS project.

## Scope
We aim to reach a global consensus among patients, parents, researchers and physicians on a minimal set of core outcomes that should be measured and reported in all future clinical trials investigating any type of treatment for acute uncomplicated appendicitis in children, including surgical treatment, NOT or other treatments aimed at curing appendicitis.

## Methods
In the development of this protocol, we adhere to the Core Outcome Set-STAndards for Development (COS-STAD) recommendations[17] and the Core Outcome Measures in Effectiveness Trials (COMET) handbook.[18] The completed COS-STAD checklist can be found in online supplementary S1. The final COS will be reported in accordance with the COS-STAndards for Reporting statement.[19] Involvement of patients and the public will be described using the Guidance for Reporting on Involvement of Patients and Public 2 reporting checklist.[20 21]

## Study design
The paediatric appendicitis COS (PA-COS) development will consist of three phases: (1) an update of the 2015 systematic review on outcomes reported in uncomplicated paediatric appendicitis research,[13] aiming to identify any additional outcomes used in trials that were published since the previous systematic review; (2) a three-step Delphi study to identify a set of core outcomes from those selected in the literature review. Development of the Delphi is performed according to the checklist by Sinha *et al*[22] on the design and reporting of Delphi studies concerning COS selection; and (3) an expert panel meeting including physicians, researchers and children/parent representatives in order to ratify the final COS. Children and young people will be involved through their parents during phase 2 and will be engaged directly using a customised approach.

## Steering committee
An international steering committee has been established and consists of the following; the authors, a parent/patient representative of the Dutch Foundation: 'Children and Hospital' and the lead local investigator of each participating centre (PA-COS development group). The steering committee will agree on the final version of the protocol at the start of the project and will provide input throughout the duration of the project. The steering committee members will also be involved in the development of the final COS. Within the steering committee, a smaller study management group has been appointed which will convene during regular (videoconference) meetings.

## Systematic review: treatment outcomes
Hall *et al* performed a systematic review of RCTs and meta-analyses reporting treatment outcomes of children with appendicitis up to April 2014.[13] They reported 115 unique outcomes which were collapsed into a total of 38 standardised outcome terms. We will update the systematic review in order to identify any new unique outcomes in clinical trials or systematic reviews. All RCTs and systematic reviews/meta-analyses reporting treatment outcomes of acute uncomplicated appendicitis in children (<18 years of age) published between 1 January 2014 and 23 November 2017 will be included. The final review will follow the Preferred Reporting Items for Systematic Reviews and Meta-Analyses reporting guideline.[23] We will search the Cochrane Central Register of Controlled Trials, MEDLINE and EMBASE with the help of a clinical librarian. Additional information on the search strategy/study selection and data extraction can be found in online supplementary S2. Studies reporting only outcomes of treatment in complex or complicated appendicitis (eg, gangrenous or perforated appendicitis, appendiceal mass and appendiceal abscess) will be excluded.

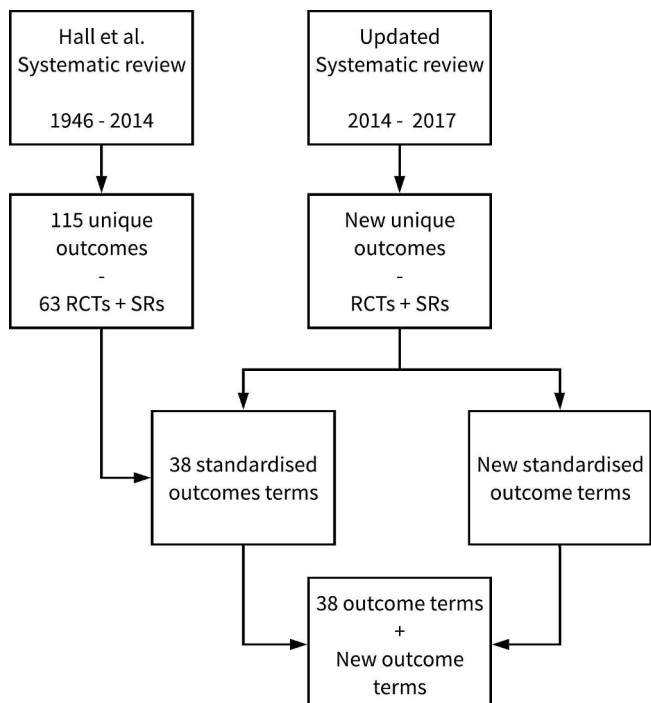

**Figure 1** Schematic depiction of outcome term selection from SRs. RCT, randomised controlled trial; SR, systematic review.

After data extraction, a meeting of the study management group (including NJ Hall) will be held to discuss potential similarities between the outcomes from the 2015 systematic review from Hall *et al.*[13] New unique outcomes will be discussed within the group in order to assign an appropriate standardised outcome term. If these outcomes do not match any of the original 38 outcome terms, a new term will be assigned; this methodology is illustrated in figure 1. The new and original outcome terms will be mapped to four core areas (death, life impact, resource use and pathophysiological manifestations) in accordance with the methods from the OMERACT Filter V.2.0.[24] Although Hall *et al*[13] chose to list the adverse events as a separate core area, we will reclassify these outcome terms to one of the four core areas (table 1). Adverse events of treatment will, however, be labelled separately, as the OMERACT filter suggests.[24] A meeting of the study management group will be held

**Table 1** Outcome core areas

| Core area | Example(s) |
| --- | --- |
| Life impact | Quality of life and loss of ability to work |
| Resource use | Length of hospital stay, healthcare costs and societal cost |
| Pathophysiological manifestations | Biochemical parameters, organ function and (ir)reversible manifestations (complications and pathology results) |
| Death | Death |

to discuss potential similarities between outcomes and to assign appropriate common outcome terms for corresponding outcomes. Outcomes that are found only once and are not generalisable can be excluded (eg, the width of lateral thermal damage of the mesoappendix after appendectomy). Grouping the outcomes under a common outcome term aims to arrive at a manageable and cohesive list of outcomes that is appropriate as a basis for the Delphi questionnaire.

### Stakeholders and recruitment
#### Children and young people
This study includes children and young people (5–18 years) who have been treated for acute uncomplicated appendicitis in the preceding 24 months, either with initial NOT or with surgery. Children less than 5 years old are excluded as different outcomes might be appropriate in this very young age group. Also, uncomplicated appendicitis is much less common in young children than in older children. Furthermore, there are no studies in which children below the age of 5 are treated non-operatively. Children will be engaged indirectly as we will urge parents to discuss the answers they provide with their child while filling out the Delphi questionnaire. Young people will be engaged directly through a customised face-to-face approach in selected countries. For the invited children, considering the complexity of the subject and methodology, age is limited to 12–18 years.

#### Parents
Parents of children and young people (5–18 years) treated for acute uncomplicated appendicitis either with initial NOT or with surgery in the preceding 24 months or during the initial phase of the study were included. Parents will be asked to discuss the answers they provide with their child while filling out the Delphi questionnaire. Parents will be invited to participate by their child's treating physician or their designate in each participating country/hospital. Participants will be identified retrospectively by contacting patients who were treated in the past 24 months or prospectively by inviting parents to participate after their child has completed the treatment.

#### Surgeons
General and/or paediatric surgeons who care for children in the specified age group will be asked to participate. Surgeons will be identified and invited by the local coordinators in each participating country. These local coordinators are research groups that have previously registered a clinical trial on uncomplicated appendicitis in children. This should allow for inclusion of physicians who also have experience in research on the treatment of appendicitis.

### Participating countries and research groups
It was decided to invite research groups that are currently conducting clinical trials on the treatment of acute uncomplicated appendicitis in children. Groups were identified through www.clinicaltrials.gov by searching

(January 2017) for 'appendicitis' with an age limitation of 5–18 years. Studies with a mixed population (children and adults) were excluded. Studies that had been completed before 2014, had not been updated since 2015 or with incomplete registrations were excluded. We found 111 trials, of which 12 trials assessed the treatment of uncomplicated appendicitis in children. Groups from the Netherlands, USA, Canada, Australia, Sweden, Finland, UK, France, Italy, Israel, Japan, Singapore and Malaysia were identified. Some trials included hospitals from multiple countries.

## Sample size

There is no rationale for determining the number of respondents to invite for a Delphi study.[18] A minimum of seven respondents per stakeholder group is suggested to have a large enough group to allow for a consensus process.[25] Taking into account that only some invited participants will register for the Delphi and not all respondents will complete all rounds of the Delphi study (attrition), a minimum of 40 respondents per stakeholder group per country will be invited. There will be no maximum. In case the number of respondents per country is significantly higher than that of other countries, we will consider a weightage per country in the analyses. We anticipate that this sample will be large enough to reflect all relevant opinions.

## Delphi study

### International online Delphi study

The Delphi method is an effective tool for reaching consensus in a large group without the need for face-to-face contact.[26] The use of sequential questionnaires, which are answered anonymously by stakeholders, is an established method for reaching consensus in a group of experts.[22] Questionnaires will be sent using DelphiManager,[27] a web-based system designed for Delphi studies. The questionnaires will be open simultaneously to all respondents of the participating countries. After each round, the aggregated responses of all participants are shared anonymously in accordance with the Delphi principle.

The list of outcomes from the systematic review will be formatted into questions, accompanied by an extensive plain language summary per outcome, including figures if appropriate. The Delphi questionnaire will originally be formulated in English and will be translated if required. Translation will only be performed by native-speaking professionals.

Participants will be asked to score the importance of each outcome using a 1-point to 9-point Likert scale as recommended by the Grading of Recommendations Assessment, Development and Evaluation working group[28] and COMET initiative.[18] A score of 7–9 indicates a critical outcome for assessing the effect of a treatment, 4–6 indicates an outcome that is important but not critical and 1–3 indicates an outcome with low importance for assessing the treatment effect. It will also be possible

to select an 'unable to score' option, which is especially of importance in case parents do not feel equipped to score certain outcomes. The questionnaires, including the plain language summaries, will be piloted by a group of laypersons (n=10) to check for ambiguity and readability.

### Delphi round 1

Participants will be divided into two stakeholder groups: parents (with their children) and surgeons. Parents will be asked to discuss the answers they provide with their child while filling out the Delphi questionnaire. Baseline characteristics (age and country) will be ascertained. Parents will be asked if their child was treated with non-operative or operative treatment, time between registration and the first diagnosis of appendicitis, and if their treatment was with or without complications. They will also be asked whether they will be answering the Delphi together with their child. Surgeons will be asked their specialty (paediatric, general, abdominal or other), workplace (academic, teaching hospital or non-teaching hospital), experience with NOT and experience in research regarding appendicitis in children.

All participants will be asked to score all previously identified outcomes according to their perceived importance for assessing the treatment effect. In the first round, there will be an option to suggest additional outcomes not yet listed.

Participants will have between 4 and 8 weeks to complete each round, depending on the response rate. During that time, they will receive a reminder email every 2 weeks as long as they have not replied to the questionnaire.

### Delphi round 1: analysis

Results will be analysed *by stakeholder group and for all participants* using descriptive statistics. Outcomes will be analysed separately for each stakeholder group, as there is evidence that patients are likely to assign importance to outcomes differently from surgeons,[29] which has the potential to influence eventual outcome selection.

'Consensus-in' will be defined as

► Greater than 70% of participants in *both* stakeholder groups scoring the outcome as 7–9 and less than 15% in *both* stakeholder groups scoring the outcome as 1–3.

► Greater than 90% of participants within *one* stakeholder group scoring the outcome as 7–9. This implies that these outcomes are highly regarded by an individual stakeholder group and should also be included.[18]

'Consensus-out' will be defined as

► Greater than 70% of participants in *both* stakeholder groups scoring the outcomes as 1–3 and less than 15% of participants in *both* stakeholder groups scoring the outcome as 7–9. Consensus-out can only be reached when there is consensus across *both* stakeholder groups.

Outcomes that do not meet any of these criteria will be defined as 'no consensus'. A stratified analysis will

be performed to check for skewing as a result of divergent opinions from a single country, or surgeons with or without research experience.

At the end of round 1, there will be a meeting of the study management group to assess whether an alteration in the Delphi study is appropriate. If additional outcomes are suggested by Delphi participants, each outcome will be assessed by the study management group to determine whether it is indeed new and to which category it should be classified. Wording of the Delphi questionnaire will be adjusted if misinterpretation is suspected.

### Delphi rounds 2 and 3

All participants who complete the previous round will be asked to participate in the next round. Only outcomes that have not yet been defined as consensus-in or consensus-out during the previous round will be presented in the following rounds to *all* participants. Outcomes for which there was only consensus-in within a single stakeholder group will still be presented to the other stakeholder group to evaluate whether consensus can be achieved in both stakeholder groups. An overview of included and excluded outcomes will be available. The outcomes for which there is no consensus and the newly suggested outcomes from the previous rounds will be presented with the participants' individual scores and the median scores from each stakeholder group combined with a histogram showing the scoring distribution. Participants will be asked to score all remaining outcomes in the same manner as in round 1.

### Delphi rounds 2 and 3 analysis

Results will be analysed per stakeholder group and for all participants using descriptive statistics, including a stratified analysis. The same definitions for consensus in/out as in the first Delphi round are upheld. After the second round, there will be a meeting of the study management group to assess the need for alterations in the Delphi study and to decide whether or not to proceed with a third Delphi round, assuming consensus between *both* stakeholder groups on more than 80% of the outcomes and more than five outcomes with consensus in. To give an estimate of the degree of agreement between respondents, the width of the IQR of the median ranking score will be calculated, potentially ranging from 0, meaning complete agreement, to 8, meaning least possible agreement. This will be calculated for both the individual stakeholder groups and the entire group of respondents after the final round.

### Face-to-face engagement of young people

We wish to check for discrepancies of opinion between parents answering the Delphi together with their child and children who are interviewed directly. For this, a form of in-person interaction will be organised with young people (12–18 years) who have been treated for appendicitis. They will be asked to comment on the preliminary COS selection established at the end of the Delphi study and to suggest additional outcomes and comment on outcomes that did not make the preliminary COS selection. This will either be done by a short, face-to-face, one-round questionnaire involving only outcomes relevant to children/young people, or in the form of a small consensus meeting (prioritisation meeting) before finalising the definitive COS. Doing this type of research requires experienced interviewers and resources. That is why the face-to-face engagement will take place only in selected countries; however, we will aim to involve as many countries as feasible. Separate ethical board approval will be obtained as appropriate.

### Consensus discussion

If adequate consensus (we aim to achieve consensus on at least one outcome per OMERACT core area) is reached in the Delphi study, we will organise a face-to-face expert panel meeting with selected individuals with the purpose to ratify a pragmatic and well-defined set of outcomes. A secondary aim of this meeting is to enhance support and implementation of the final COS.

The meeting will be held at an international conference for paediatric surgery. Through purposive sampling, approximately 30 'experts' from across all stakeholder groups, including physicians, researchers and children/parent representatives, will be invited to participate in a face-to-face meeting with the steering committee. Journal editors and healthcare commissioners will also be invited to attend in an observational capacity with the purpose of promoting implementation and to provide comments on the final list of outcomes.

In the event that adequate consensus cannot be reached in the Delphi process, we will organise a formal face-to-face consensus meeting or teleconference. In that case, we will select an appropriate representation of all stakeholder groups from the panel members who participated in the Delphi study.

### Final COS development

The goal is to achieve a pragmatic COS that is applicable and feasible for all future trials that evaluate the treatment of uncomplicated appendicitis in children. There is no recommended maximum number of outcomes that should be included in a COS. However, if the final COS includes too many outcomes, the COS would not be feasible to use in practice. To achieve the goal of a pragmatic COS, we aim to arrive at a maximum of 10 outcomes, the same maximum number as the UK COS protocol specifies.[15] As a minimum, we aim to have at least one outcome per core area. If the number of outcomes for which consensus is achieved greatly exceeds 10 outcomes, the outcomes with the highest level of consensus will be considered part of the suggested COS. However, we will report all outcomes for which consensus is achieved. The highest level of consensus depends on whether there is consensus in both stakeholder groups, the median score that was appointed to the outcome and the IQR of the median score as an estimate of the degree of consensus.

Only outcomes for which consensus is reached internationally will be selected. To test for country bias, stratified analyses of the Delphi results will be performed. The results from the face-to-face engagement of young people will be taken into account for the final COS selection and will be reported separately. If there is no consensus between patients, parents and healthcare professionals, an outcome can still be selected if there is clear consensus within a single stakeholder group. These will be reported separately. The final COS will be categorised according to the four core areas of the OMERACT filter.[24] We will also annotate the outcomes according to the recently published outcome taxonomy to maximise future data harmonisation.[30]

### Patient and public involvement

Patient involvement is at the core of this study design. As we will directly be be asking parents and patients, with experience in having uncomplicated appendicitis, what outcomes they feel should be part of future research. To ensure our design is appropriate for parents and children, we have involved the Dutch child and parents representation group as part of the steering committee. In that capacity, they provide input on the protocol and the study. To make sure the Delphi questionnaire is understandable and has no ambiguities, it is checked by a group of laypersons before the start of the Delphi study. Part of the Delphi study is giving feedback to all its participants after each round; this will also be done with the final study results.

### Ethics and dissemination

#### Ethics

The medical research ethics committee of the Academic Medical Center Amsterdam confirmed that the Dutch Medical Research Involving Human Subjects Act does not apply to this study and that complete approval of this study by the committee is not required. Each participating country/research group will be asked to obtain ethics board approval or confirm that ethics board approval is not required. Electronic informed consent will be obtained from all participants. The face-to-face engagement of young people (12–18 years) will take place in selected countries, and separate ethics board approval will be obtained, as appropriate.

#### Data collection and confidentiality

All data will be handled confidentially and in accordance with the Dutch Personal Data Protection Act and the European General Data Protection Regulation. Delphi-Manager[27] will be used for the online questionnaire. After obtaining informed consent from all participants, only limited identifying information (name and email) will be ascertained during registration. This information will be stored separately from the answers given in the questionnaire and will be used only for the purpose of direct feedback and reminder emails. Access to personally identifiable data will be strictly limited.

### Study status and dissemination

In the first quarter (Q1) of 2018, the following 13 countries were invited to participate in the project: the Netherlands, USA, Canada, Australia, Sweden, Finland, UK, France, Italy, Israel, Japan, Singapore and Malaysia. Ten countries replied; Italy, Israel and Japan did not. In Q1 of 2018, the systematic review was finished. In the second quarter of 2018, the Delphi questionnaire was developed and piloted. In the third quarter of 2018, all materials were translated. Between the fourth quarter of 2018 and Q1 of 2019, institutional review board applications were submitted in 10 countries and 15 participating centres. The anticipated start of the online Delphi study is May 2019. We anticipate to have the final COS ready by Q1 of 2020. Dissemination of the results will be accomplished by publication in an international peer-reviewed scientific journal and by presentations at (international) conferences. By involving the majority of the principal investigators who are currently involved in research on uncomplicated appendicitis in children, we aim to optimise uptake of the final COS. By involving journal editors and healthcare commissioners in the face-to-face consensus discussion, we aim to ultimately have the COS introduced as a requirement in future outcome reporting on the treatment of uncomplicated appendicitis in children. We will also actively send out the final COS to relevant journal editors and funding bodies to promote uptake in future research.

## DISCUSSION

### Strengths and limitations of this study

#### Outcomes selection

The selection of potential outcomes will be done systematically and will provide a selection for the first Delphi questionnaire that reflects most issues pertinent to the treatment of uncomplicated appendicitis. By including systematic reviews/meta-analyses that also report on non-comparative studies, we expect to identify all reported treatment outcomes, including those from the relatively new field of NOT for uncomplicated appendicitis.

To be able to arrive at a manageable list of outcomes that is appropriate for a Delphi study, the number of outcome terms needs to be somewhat limited. In order to achieve this, the outcomes derived from our systematic review will be merged in case of similarity. If outcomes are not generalisable and are reported only once, they will be excluded. This will be proposed and prepared by two independent reviewers and discussed in the study management group. However, the merging of outcomes will inevitably lead to some loss of detail.

#### Global consensus

In order to reflect the views of different stakeholders, a variety of groups will be part of the development of this COS. This is the case not only on a national level but also on an international level, related to, for example, differences between countries in resources, treatment

practices for acute uncomplicated appendicitis and cultural differences. For example, there is a large difference with regard to the standard length of hospital stay after an appendectomy for uncomplicated appendicitis. In the USA, much effort is devoted to reduce the number of admission days; in the UK, there is only limited attention for the duration of admission, and for instance, in Japan, an admission for 5 days is not uncommon. We can expect that these kinds of differences result in different opinions regarding the COS. By also involving patients and parents from the participating countries, we hope to correct for these differences.[31] In conjunction with the UK PA-COS research group, we decided that an international validation of the UK COS would not give the depth of information and would not allow for consensus formation on all possible outcomes, which we feel is appropriate, considering the previously mentioned significant differences between countries. Involving members from different countries should lead not only to the development of a COS that reflects the opinions of the international community but also to an internationally applicable 'minimal' COS. However, selecting the participating countries on the basis of their involvement in research on appendicitis in children is a limitation. This choice was made on the basis of feasibility. Researchers in the field of uncomplicated appendicitis have an interest in the development of a COS and have the network to help carry out the Delphi study. With our current selection, we will still have participants from four different continents. Our method of country selection has another advantage. Since NOT is an important research subject in childhood appendicitis, we aim to include surgeons and parents who have experience in that field. As NOT is still experimental in most countries, we also need surgeons and patients who have been involved in such research.

### Limited face-to-face consensus

If consensus is reached in the Delphi study, we will not be organising a formal consensus meeting. The Delphi method can be used for reaching consensus in a group of respondents without the need for face-to-face contact. There is a risk of bias if a face-to-face consensus meeting leads to selection of only participants who are able to attend the meeting, which is especially a problem in a global consensus procedure. There are also problems regarding language barriers in an international consensus meeting. To check for interpretation errors in the Delphi method and to ensure a pragmatic and well-defined set of outcomes, the results of the Delphi study will be discussed in an (international) expert meeting. However, the influence of this meeting on which outcomes are selected for the final COS is very limited, as this selection is primarily made in the Delphi study.

### Involving parents and their children

Involving patients in COS development has recently become common practice with 88% (n=112 as of 12 April 2016) of ongoing COS development studies doing so.[18] Involving

patients as participants seems imperative as patients may select different outcomes, compared with physicians.[16] For this protocol, we performed a scoping review (unpublished data, Knaapen M, 2018) that found 12 studies that directly engaged children in COS development, either as part of the advisory group or the steering committee, or as a stakeholder group in the Delphi,[15 22] focus groups,[32] interviews[33] or as a part of the consensus meeting.[34] Attempts to engage children and young people in an online Delphi questionnaire have proven to be difficult. In the UK COS for uncomplicated appendicitis, there were substantial difficulties with retaining young people in the consecutive rounds of the Delphi questionnaire, despite extensive efforts to optimise the methodology to appeal to children and young people, including preliminary semistructured interviews on the subject, pretesting of the Delphi survey by young people and children,[15] and video animations explaining the need for a COS. However, parent participation showed more promising results. Consequently, to safeguard the input of children/young people, the Delphi questionnaire for this study will be developed to be completed by parents with input from their children (5–18 years) whenever possible. In order to ensure that there are no large discrepancies between the opinions of parents together with their child, and with children without their parents, we will organise a form of in-person interaction with young people (12–18 years) who have been treated for appendicitis. Involving children/young people in COS development is a subject of interest in many ongoing COS development projects. As the search for the optimal approach to engage young people is ongoing, we have not yet selected a final methodology. Two members of the study management group are currently involved in a group that is developing such methodology in consultation with young people themselves. We will update our protocol as soon as we settle on a methodology before starting the face-to-face engagement. The updated protocol will be published on an online, open-source format (via the Open Science Framework).

A limitation is that due to the international nature of our study, it will not be feasible to engage children directly in all the participating countries. That is why the face-to-face engagement will take place in selected countries.

### Other stakeholders

After careful consideration and consultation with the participating countries, it was decided not to include paediatricians, general practitioners, nurses or emergency medicine physicians. Although all these specialists play an intricate role in the diagnosis and care for children with appendicitis, they do not make the final decision regarding treatment or its provision. However, we will however, depending on the organisation of the healthcare system in each country, ask these stakeholders to comment on the final COS in order to ensure that essential outcomes are not missed. Since almost all research regarding treatment of paediatric uncomplicated appendicitis is initiated by (paediatric) surgeons, it was decided that researchers will not be included as a

separate individual stakeholder group. However, involvement in research will be registered. While their opinion is vital to the development of a COS, it is likely researchers will be well represented in the (paediatric) surgeon stakeholder group. A stratified analysis will be performed to check for skewing of the results by surgeons involved in research. It was also decided not to include journal editors or healthcare commissioners. Even though their opinion is of great importance especially regarding implementation, it was determined that their opinion is not essential in establishing the outcomes selected for the COS. Also, there is much variability between countries regarding the role of these stakeholders, which would lead to major challenges regarding Delphi analyses of such a small stakeholder group. However, to enhance implementation and because of their expertise on the use of COSs, representatives of these stakeholder groups will be asked to attend the final consensus discussion.

## Outcome measures

This study will not answer the question on how to measure the outcomes that are included in the final COS or at what time point the outcomes should be measured. However, we will attempt to come to a clear definition of each outcome. We expect that further research will be necessary to answer the question of timing and how to measure the outcomes. We will advise on this subject in the final report.

**Author affiliations**
[1]Paediatric Surgical Centre, Emma Children's Hospital, Amsterdam UMC, University of Amsterdam, Vrije Universiteit, Amsterdam, The Netherlands
[2]University Surgery Unit, University of Southampton Faculty of Medicine, Southampton, UK
[3]Paediatric Clinical Research Office, Emma Children's Hospital, Amsterdam UMC, University of Amsterdam, Amsterdam, The Netherlands
[4]Child Health Evaluative Sciences, Sick Kids, Toronto, Ontario, Canada
[5]Department of Paediatrics, The Hospital for Sick Children, University of Toronto, Toronto, Ontario, Canada

**Acknowledgements** We acknowledge Hester Rippen as the representative of the Dutch Foundation Children and Hospital for her advice and support in drafting the protocol.

**Collaborators** Paediatric Appendicitis COS development group: on behalf of the Paediatric Appendicitis COS development group: P C Minneci, J F Svensson, F I Luks, S D St. Peter, O Abbo, A P Arnaud, S Adams;s, S A Nah, E D Skarsgard, A Pierro, A Zani, S Emil, R Keijzer, J S Suominen and DA Aziz. The paediatric surgery departments of the following hospitals have initiated the COS project and will contribute by recruiting participants: Nationwide Children's Hospital, Columbus, OH, USA; Hasbro Children's Hospital and Alpert Medical School of Brown University, Providence, RI, USA; Children's Mercy Hospital, Kansas City, MO, USA; Karolinska University Hospital, Stockholm, Sweden; Southampton General Hospital, Southampton, UK; Hôpital des Enfants, Centre Hospitalier Universitaire Toulouse, Toulouse, France; Hôpital Femme-Enfant, University Hospital, CHU Rennes, Rennes, France; Sydney Children's Hospital, Randwick NSW, Australia; KK Women's and Children's Hospital, Singapore; BC Children's Hospital, Vancouver, BC, Canada; The Hospital for Sick Children, Toronto, ON, Canada; Montreal Children's Hospital, Montreal, QC, Canada; Emma Children's Hospital, Amsterdam UMC, University of Amsterdam, Vrije Universiteit, Amsterdam, the Netherlands; Helsinki Children's Hospital, Helsinki, Finland; and Universiti Kebangsaan Malaysia Medical Centre, Kuala Lumpur, Malaysia.

**Contributors** All authors contributed to the design of this protocol. MK, NJH JHvdL, RB and RRG initiated the project. The protocol was drafted by MK and was refined by NJH, JHvdL, RB, MO, NJB, EWEvH and RRG. Statistical advice was provided by JHvdL. MK was responsible for drafting the manuscript. All authors contributed to the manuscript and read and approved the final manuscript. The Paediatric Appendicitis COS Development Group consists of all local investigators who are responsible for translation, ethical board approval and participant recruitment. They have all read, refined and approved the final manuscript.

**Funding** The authors have not declared a specific grant for this research from any funding agency in the public, commercial or not-for-profit sectors.

**Competing interests** None declared.

**Patient consent for publication** Not required.

**Provenance and peer review** Not commissioned; externally peer reviewed.

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
