## [Reviewer comments · BMJ Open]

ARTICLE DETAILS

TITLE (PROVISIONAL)	Establishing a core outcome set for treatment of uncomplicated appendicitis in children: study protocol for an international Delphi survey.
AUTHORS	Knaapen, Max; Hall, Nigel; van der Lee, Johanna H.; Butcher, Nancy; Offringa, Martin; Van Heurn, Ernst; Bakx, Roel; Gorter, Ramon; + On behalf of the paediatric, appendicitis COS development group

VERSION 1 - REVIEW

REVIEWER	Francois Dos Santos Imperial College London, London, United Kingdom
REVIEW RETURNED	18-Jan-2019

GENERAL COMMENTS	Introduction - Well written- Good definition of scope Methods - Outcomes included up to 2014: The 115 unique outcomes were 150 collapsed into a total of 38 standardized outcome terms - explain if will use the 38 outcomes already "collapsed" by someone else or if they will start with the original 115, add the new ones from the updates SR 2017 and then organise them consulting with the steering committee. A diagram would probably be helpful.- "Children will be engaged (indirectly) through their parents in the Delphi questionnaire, and directly through a customized face-to-face approach in selected countries" It's not clear exactly what is going to happen. What does indirectly through their parents mean? Is it the what the parents think is important for their children or will the children actually have a say? Also if there is the opportunity to do face-to-face why only in selected countries? I would prefer the methods to be the same across countries but understand if not possible.- Stakeholders and Recruitment Have you considered adding Nurses as a stakeholder group. They will be a valuable asset as they are looking after these children and in close contact with the children and the parents. They might have interesting views on NOT.- Delphi "extensive additional information per outcome" What does this mean? Is it a plain language summary? Will this go on the
---

	questionnaire itself or as a different doc. Which platform is going to be used to define each outcome? - Reminder emails two reminder emails in 8 week period doesn't seem like much. I would send at least every two weeks - just a suggestion from personal experience. - Final COS "If consensus is reached on more than 10 outcomes, the 10 outcomes with the highest level of consensus will be considered part of the suggested COS." I agree that to be feasible a COS should not have many outcomes. However, limiting it to 10 seems to be a random decision not base don evidence. Why do the authors want to limit to 10 and not 11? Systematic review The authors specify this is a protocol for a SR and the Dlephi. I get the impression that even though the SR methods are described briefly, this is mainly a protocol for the Delphi. A protocol for the SR would be an entire different document - please see PROSPERO guide (https://www.crd.york.ac.uk/prospero/). I would suggest removing systemaic review from the title. Overall very well written, well structured protocol following current guidelines for the development of COS
--	---

REVIEWER	Rebecca Fish University of Manchester, United Kingdom
REVIEW RETURNED	04-Feb-2019

GENERAL COMMENTS	In general this is a well written manuscript and describes a well thought through study protocol. The justification for choice of methodological approach is clearly described in most cases and the approach is consistent with the latest methodological recommendations/guidelines in the field (COMET, OMERACT). Of concern is the apparent overlap with the UK childhood appendicitis core outcome set. The authors reference the UK COS, stating "The development of this protocol and the international COS is being performed in conjunction with the UK research group" [p3, line 104-105]. However, no further detail is given. Some COS developers take the view that if a well-developed COS already exists, that COS should be validated for use in other (international) populations, rather than developing a new COS from scratch (McNair 2017, MacLennan 2017). I would therefore like to hear in more detail how authors are working in conjunction with the UK group to prevent duplication of effort. A secondary related concern is the degree to which this international core outcome set will be truly international. If the only difference between the UK COS and the current project is the international participation, I would be concerned that there is not enough international participation guaranteed in this project. More detail on which centres will be participating is needed: Of the groups identified [p7, lines 207-216], are all being invited? Have any agreed to participate already? Have any been invited to the steering committee already? Which 'selected countries' [p10, line
---

	324] will be taking part in the face-to-face engagement of young people? Finally, the authors describe mapping outcomes into four core areas as described in the OMERACT filter. More recently, the COMET group has published a recommended outcome taxonomy (Dodd et al 2018) which is being widely adopted for use in COS development and by Cochrane. The authors should consider this taxonomy to classify their outcomes to maximise data harmonisation.
--	---

REVIEWER	Sarah Gorst University of Liverpool, UK
REVIEW RETURNED	08-Feb-2019

GENERAL COMMENTS	This a well written protocol, which clearly explains how the core outcome set for uncomplicated appendicitis in children will be developed. It is great to see that the authors have adhered to the COS-STAD recommendations. I just have a couple of minor comments, which are generally for clarification. The journal recommends that study protocols should include the dates of the study. The authors provide the date that the study was registered with the COMET Initiative and the dates of the systematic review search, but the dates for the Delphi survey and potential consensus meeting are not provided. On page 4, line 100-102, the authors refer to a study protocol for an appendicitis COS being developed in the UK and go on to state that the current study will now develop an international COS "to overcome any limitations of a COS focused on UK-specific surgical practice". Can the authors describe what these potential limitations would be? On page 4, line 106-107, the authors state "Parent and patient representation in the development of this protocol was provided through the Dutch patient and parent Foundation". Can the authors describe specifically how patients and parents were involved in the development of this protocol, what was their input? On page 13, lines 376-383, the authors refer to dissemination of the COS and whilst it is great to hear that the authors "aim to ultimately have the COS introduced as a requirement in future outcome reporting on the treatment of uncomplicated appendicitis in children" by involving journal editors and healthcare commissioners in the consensus discussion, it would be interesting to read about your specific plans for implementation plans following the development of the COS?
--

VERSION 1 – AUTHOR RESPONSE

Reviewer 1:

1: "Outcomes included up to 2014: The 115 unique outcomes were 150 collapsed into a total of 38 standardized outcome terms - explain if will use the 38 outcomes already "collapsed" by someone else or if they will start with the original 115, add the new ones from the updates SR 2017 and then organise them consulting with the steering committee. A diagram would probably be helpful"

Response 1: Thank you for your comment. We agree that the drafting of the new outcome terms can be described more clearly. We have altered the text accordingly and added a diagram as suggested. Named Figure 1. which we have uploaded.

Page 5 Line 156: "Hall et al. performed a systematic review of RCTs and meta-analyses reporting treatment outcomes in children with appendicitis up to April 2014[13]. They found a 115 unique outcomes which they collapsed into a total of 38 standardized outcome terms. We will update the systematic review to identify any new unique outcomes in clinical trials or systematic reviews."

"After data extraction, a meeting of the study management group (including NJ Hall) will be held to discuss potential similarities between the outcomes from the 2015 systematic review from Hall et al.[13]. New unique outcomes will be discussed within the group in order to assign an appropriate standardized outcome term. If these outcomes do not match any of the original 38 outcome terms a new term will be assigned, figure 1. illustrates this methodology."

2: "Children will be engaged (indirectly) through their parents in the Delphi questionnaire, and directly through a customized face-to-face approach in selected countries" It's not clear exactly what is going to happen. What does indirectly through their parents mean? Is it the what the parents think is important for their children or will the children actually have a say? Also if there is the opportunity to do face-to-face why only in selected countries? I would prefer the methods to be the same across countries but understand if not possible."

Response 2: The experience with involvement of children in core outcome set development is relatively limited. In a scoping review we performed for this protocol we found 12 studies that involved children. This was done either through (online)questionnaires, semi-structured interviews, focus groups and/or consensus meetings. The UK pediatric appendicitis COS group tried involving children through an online Delphi. However, they experienced substantial difficulties with retaining young people in the consecutive rounds of the Delphi questionnaire. Despite extensive efforts to optimize the methodology to appeal to children and young people, including: preliminary semi-structured interviews. That is why we chose, in conjunction with the UK group, to ask parents to discuss the answers they provide with their child whilst filling out the Delphi questionnaire. We will however ask parents during the registration if they will involve their children, however, there is no way for us to verify if children are actually being involved. That is why we consider it an indirect engagement of children in the Delphi. We explain these principles on page 9 line 267 and in the discussion Page 14 line 483

With regard to the face-to-face involvement of young people. There is very limited experience in using child specific methodology for COS development, and so far there is no preferred methodology. For the methodology described on page 10 line 331 we consulted the KLIK project a group of the psychosocial department at the Emma Children's Hospital AMC that specializes in developing Patient Reported Outcomes that children can report themselves. Furthermore, two authors (Knaapen and Hall) are part of the COSMIC (Core Outcome Set Methodology in Children) group. A group, related to COMET, that is currently in the process of developing methodology to involve children in COS development, also by consulting young people themselves. We are hoping to have more evidence regarding the optimal approach to engage young people by the time we start the face-to-face

involvement. That is why we have not yet selected a methodology, we will update our protocol as soon as we settle on the final methodology. This updated protocol will be published in an open source network as stated in the discussion Page 15 line 512. Doing the face-to-face engagement in all the participating countries would be ideal. However, this requires resources and experience with doing this type of research in children. That is why we will aim to involve as many countries as feasible. However, this will depend on the final methodology and the extent of discrepancies between the Delphi results and face-to-face results from the first pilot(s) face-to-face meetings.

We have adjusted the manuscript to more clearly reflect the above mentioned considerations. We hope this answers the questions you raised in your review.

Page 6 Line 195: "Children will be engaged indirectly as we will urge parents to discuss the answers they provide with their child whilst filling out the Delphi questionnaire. Young people will be engaged directly through a customized face-to-face approach in selected countries."

Page 8 Line 269: "Parents will be asked if their child was treated with non-operative or operative treatment, time between registration and the first diagnosis of appendicitis and if their treatment was with or without complications. And also if they will be answering the Delphi whilst consulting their child."

Page 10 Line 340: "Doing this type of research requires experienced interviewers and resources. That is why the face-to-face engagement will only take place in selected countries, however, we will aim to involve as many countries as feasible. Separate ethical board approval will be obtained as appropriate."

Page 14 Line 485: "Involving patients as participants seems imperative as patients may identify different outcomes that should be measured, compared with physicians[16]. For this protocol we performed a scoping review [unpublished work] that found 12 studies that directly engaged children in COS development. Either as part of the advisory group or the steering committee, or as a stakeholder group in the Delphi[15,22], focus groups[31], interviews[32] or as a part of the consensus meeting[33]."

Page 15 Line 505: "As the search for the optimal approach to engage young people is ongoing we have not yet selected a final methodology. Two members of the study management group are currently involved in group that is developing such methodology in consultation with young people themselves. We will update our protocol as soon as we settle on a methodology before starting the face-to-face engagement. The updated protocol will be published on an online, open source format (via the Open Science Framework)."

3: "Have you considered adding Nurses as a stakeholder group. They will be a valuable asset as they are looking after these children and in close contact with the children and the parents. They might have interesting views on NOT."

Response 3: We did briefly consider nurses as a relevant stakeholder group. However, the same can be said for pediatricians, general practitioners, or emergency medicine physicians. All these healthcare professionals play an important role in the care for these children but do not make the final decisions regarding treatment options. We will however, the same as for pediatricians, general practitioners and emergency medicine physicians, ask nurses to comment on the final COS in order to ensure that essential outcomes are not missed. We added them as potential stakeholders to the discussion.

Page 15 Line 518: "After careful consideration and consultation with the participating countries, it was decided not to include paediatricians, general practitioners, nurses or emergency medicine physicians. Although all these specialists play an intricate role in the diagnosis and care for children with appendicitis, they do not make the final decision regarding treatment or its provision. We will

however, depending on the organisation of the healthcare system in each country, ask these stakeholders to comment on the final COS in order to ensure that essential outcomes are not missed.”

4: “extensive additional information per outcome” What does this mean? Is it a plain language summary? Will this go on the

questionnaire itself or as a different doc. Which platform is going to be used to define each outcome?”

Response 4: The extensive information is indeed a plain language summary which accompanies the online questionnaire. There will be no separate document. The outcome definitions come from the articles in which they were originally mentioned. The plain language summary comes from online resources like patient information websites (for instance WebMD, Uptodate.com patient education etc.). The questions and plain language summaries have been tested by a group of laypersons and altered according to their comments.

We have adjusted the manuscript to more clearly reflect these steps. We hope this answers the questions raised in your review.

Page 8 Line 251: “The list of outcomes from the systematic review will be formatted into questions accompanied by an extensive plain language summary per outcome, including figures if appropriate.”

Page 8 Line 262: “The questionnaires including the plain language summaries will be piloted by a group of laypersons (n=10) to check for ambiguity and readability.”

5: “two reminder emails in 8 week period doesn't seem like much. I would send at least every two weeks - just a suggestion from personal experience.”

Response 5: Thank you for this practical tip! We adjusted the protocol and text accordingly.

Page 9 Line 280: “In that time they will receive a reminder email every two weeks as long as they have not replied to the questionnaire.”

6: “If consensus is reached on more than 10 outcomes, the 10 outcomes with the highest level of consensus will be considered part of the suggested COS.” I agree that to be feasible a COS should not have many outcomes. However, limiting it to 10 seems to be a random decision not based on evidence. Why do the authors want to limit to 10 and not 11?”

Response 6: We absolutely agree that this cut-off is not evidence based. This point was discussed in the steering committee and it was decided to use the same cut-off as the UK COS protocol. Because we agree that 10 is a random number we have altered the text Changing it to; if the number of outcomes greatly exceeds 10. In the case the number greatly exceeds 10 we will consider the 10 outcomes with the highest level of consensus part of the suggested COS, for feasibility's sake. We will however report all outcomes for which consensus is achieved. We hope this answers the questions raised in your review.

Page 11 Line 364: “There is no recommended maximum number of outcomes that should be included in a COS. However, if the final COS includes too many outcomes, the COS would not be feasible to use in practice. To achieve the goal of a pragmatic COS we aim to arrive at a maximum of 10 outcomes, the same maximum number as the UK COS protocol specifies[15]. As a minimum, we aim to have at least one outcome per core area. If the number of outcomes for which consensus is achieved greatly exceeds 10 outcomes, the outcomes with the highest level of consensus will be considered part of the suggested COS. We will however report all outcomes for which consensus is achieved.”

7: "The authors specify this is a protocol for a SR and the Delphi. I get the impression that even though the SR methods are described briefly, this is mainly a protocol for the Delphi. A protocol for the SR would be an entire different document - please see PROSPERO guide (<https://www.crd.york.ac.uk/prospéro/>). I would suggest removing systematic review from the title."

Response 7: We agree that the systematic review is not the major focus of the manuscript and have removed systematic review from the title. The title is now: "Establishing a core outcome set for treatment of uncomplicated appendicitis in children: study protocol for an international Delphi survey."

Reviewer 2:

1: "Of concern is the apparent overlap with the UK childhood appendicitis core outcome set. The authors reference the UK COS, stating "The development of this protocol and the international COS is being performed in conjunction with the UK research group" [p3, line 104-105]. However, no further detail is given. Some COS developers take the view that if a well-developed COS already exists, that COS should be validated for use in other (international) populations, rather than developing a new COS from scratch (McNair 2017, MacLennan 2017). I would therefore like to hear in more detail how authors are working in conjunction with the UK group to prevent duplication of effort."

Response 1: We absolutely agree that a duplication of efforts without a clear rationale would be a waste of resources and time. NJ Hall, principal investigator of UK COS has been involved from the beginning of the protocol development and is part of the study management group. We extensively discussed the outcomes of their research in relation to our protocol and systematic review results. The UK group, as well as us, feel that a "simple" international validation of the UK COS would not give the depth of information and would not allow for consensus formation on all possible outcomes. Which we feel is appropriate as we can expect some significant differences between countries because of differences in resources, treatment standards as well as cultural differences. Also we did find a significant amount of new outcomes terms in our updated systematic review that warrants a new consensus process. We hope this answers the questions raised in your review. And we made some change in the manuscript to reflect the above mentioned.

Page 3 Line 105: "The development of this protocol and the international COS is being performed in conjunction with the UK research group. The principal investigator of UK COS (NJ Hall) has been involved from the beginning of the protocol development and is part of the study management group."

Page 14 Line 456: "In conjunction with UK paediatric appendicitis COS research group we decided that an international validation of the UK COS would not give the depth of information and would not allow for consensus formation on all possible outcomes. Which we feel is appropriate considering the before mentioned significant differences between countries."

2: "secondary related concern is the degree to which this international core outcome set will be truly international. If the only difference between the UK COS and the current project is the international participation, I would be concerned that there is not enough international participation guaranteed in this project. More detail on which centers will be participating is needed: Of the groups identified [p7, lines 207-216], are all being invited? Have any agreed to participate already? Have any been invited to the steering committee already? Which 'selected countries' [p10, line 324] will be taking part in the face-to-face engagement of young people?"

Response 2: We have added a section in the dissemination heading with regard to the study status and dissemination. This includes the countries that have agreed to participate. We will have participants from 10 countries and 15 centers in four continents. Involvement from South-America and Africa would be preferable, however, we feel this is quite a large international representation. All the

local investigators of the participating centers have provide feedback as a part of the steering committee.

With regard to the face-to-face involvement of young people. There is very limited experience in using child specific methodology for COS development. Doing the face-to-face engagement in all the participating countries would be ideal. However, this requires resources and experience with doing this type of research in children. We aim to involve as many countries as feasible, however we cannot yet answer which countries will be taking part. This will depend on the final methodology and the extent of discrepancies between the Delphi results and face-to-face results from the first pilot face-to-face meeting.

We have adjusted the manuscript to more clearly reflect the above mentioned considerations. We hope this answers the questions raised in your review.

Page 10 Line 340: “Doing this type of research requires experienced interviewers and resources. That is why the face-to-face engagement will only take place in selected countries, however, we will aim to involve as many countries as feasible. Separate ethical board approval will be obtained as appropriate.

Page 13 Line 414: “In the first quarter (Q1) of 2018 the following 13 countries were invited to participate in the project; Netherlands, USA, Canada, Australia, Sweden, Finland, UK, France, Italy, Israel, Japan, Singapore and Malaysia. Ten countries replied, Italy, Israel and Japan did not. In Q1 2018 the systematic review was finished. In Q2 2018 the Delphi questionnaire was developed and tested. In Q3 2018 all materials were translated. Between Q4 2018 and Q1 2019 IRB applications were submitted in 10 countries and 15 participating centers. The anticipated start of the online Delphi study is May 2019. We anticipate to have final COS ready by Q1 2020“

Page 15 Line 505: “As the search for the optimal approach to engage young people is ongoing we have not yet selected a final methodology. We will however update our protocol as soon as we settle on a methodology before starting the face-to-face engagement. The updated protocol will be published on an online, open source format (via the Open Science Framework).”

3: “Finally, the authors describe mapping outcomes into four core areas as described in the OMERACT filter. More recently, the COMET group has published a recommended outcome taxonomy (Dodd et al 2018) which is being widely adopted for use in COS development and by Cochrane. The authors should consider this taxonomy to classify their outcomes to maximise data harmonisation. ”

Response 3: Thank you for the suggestion. We definitely applaud initiatives to improve data harmonization. We adjusted the protocol and manuscript accordingly, and added the reference to Dodd et al.

Page 12 Line 382: “We will also annotate the outcomes according to the recently published outcome taxonomy to maximise future data harmonisation[30].”

30 Dodd S, Clarke M, Becker L, et al. A taxonomy has been developed for outcomes in medical research to help improve knowledge discovery. *J Clin Epidemiol* 2018;96:84–92.
doi:10.1016/j.jclinepi.2017.12.020

Reviewer 3:

1: “The journal recommends that study protocols should include the dates of the study. The authors provide the date that the study was registered with the COMET Initiative and the dates of the

systematic review search, but the dates for the Delphi survey and potential consensus meeting are not provided.”

Response 1: We have added a section in the dissemination heading: study status and dissemination.

Page 13 Line 414: “In the first quarter (Q1) of 2018 the following 13 countries were invited to participate in the project; Netherlands, USA, Canada, Australia, Sweden, Finland, UK, France, Italy, Israel, Japan, Singapore and Malaysia. Ten countries replied, Italy, Israel and Japan did not. In Q1 2018 the systematic review was finished. In Q2 2018 the Delphi questionnaire was developed and tested. In Q3 2018 all materials were translated. Between Q4 2018 and Q1 2019 IRB applications were submitted in 10 countries and 15 participating centers. The anticipated start of the online Delphi study is May 2019. We anticipate to have final COS ready by Q1 2020“

2: “On page 4, line 100-102, the authors refer to a study protocol for an appendicitis COS being developed in the UK and go on to state that the current study will now develop an international COS "to overcome any limitations of a COS focused on UK-specific surgical practice". Can the authors describe what these potential limitations would be?”

Response 2: As described in our discussion, page 13 line 447: differences between countries in resources, treatment practises for acute uncomplicated appendicitis, and cultural differences. For example there is a large difference with regard to the normal length of hospital stay. In the USA much effort is done to reduce the number of admission days, in the UK there is limited attention for the duration of admission and for instance in Japan it is quite normal to be admitted for 7 days for an uncomplicated appendicitis. We can expect that these kind of differences result in different opinions regarding the core outcomes set. We added how we decided in conjunction with the UK group that an international COS is warranted. Specifically why the UK-surgical practice might differ is for instance the lack of standard pre-operative imaging, the NHS healthcare system, the limited experience with non-operative treatment of appendicitis, etc. And we have added an example of the difference in treatment practice around the world.

Page 13 Line 449: “For example there is a large difference with regard to the standard length of hospital stay after an appendectomy for simple appendicitis. In the USA much effort is devoted to reduce the number of admission days, in the UK there is only limited attention for the duration of admission and for instance in Japan an admission for 5 days is not uncommon. We can expect that these kind of differences result in different opinions regarding the core outcomes set. By also involving patients and parents from the participating countries we hope to correct for these differences[31]. In conjunction with the UK paediatric appendicitis COS research group we decided that an international validation of the UK COS would not give the depth of information and would not allow for consensus formation on all possible outcomes. Which we feel is appropriate considering the before mentioned significant differences between countries.”

3: “On page 4, line 106-107, the authors state "Parent and patient representation in the development of this protocol was provided through the Dutch patient and parent Foundation". Can the authors describe specifically how patients and parents were involved in the development of this protocol, what was their input?”

Response 3: We expanded our description of the patient and parent involvement as described below. The questionnaire and plain language summary was also checked by a group of laypersons as described on page 8 line 262. We hope this adequately answers your questions.

Page 3 Line 111: “Parent and patient representation was insured through involvement of the Dutch patient and parent Foundation: “Children and Hospital”. A representative from this group provided feedback from the perspective of parents and children in several stages of the protocol development.

They will also be involved in the development of a face-to-face methodology for engaging children in this COS project. “

4: “On page 13, lines 376-383, the authors refer to dissemination of the COS and whilst it is great to hear that the authors “aim to ultimately have the COS introduced as a requirement in future outcome reporting on the treatment of uncomplicated appendicitis in children” by involving journal editors and healthcare commissioners in the consensus discussion, it would be interesting to read about your specific plans for implementation plans following the development of the COS?”

Response 4: The eventual uptake of COS is of the utmost importance, as all our efforts would be in vain if future research on the subject neglects to use the COS. Apart from the points mentioned in the dissemination paragraph, we believe that involving the major share of researchers around the world that are currently involved in pediatric appendicitis research will have a big effect on COS uptake. Implementation will also be a major point of attention for the final consensus discussion, as mentioned on page 11 line 347. Besides from involving journal editors in this discussion we will send out the final COS to all relevant journal editors and funding bodies to ensure critical review of new research. Lastly our local researchers are active members of the European Paediatric Surgeons' Association, the American and Canadian Pediatric Surgical Association and the Pacific Association of Pediatric Surgeons. They will use these involvements to actively promote to use the final COS in future research in pediatric research. We adjusted the text accordingly.

Page 13 Line 423: “By involving the majority of the principal investigators who are currently involved in research on uncomplicated appendicitis in children, we aim to optimize uptake of the final COS. By involving journal editors and healthcare commissioners in the face-to-face consensus discussion, we aim to ultimately have the COS introduced as a requirement in future outcome reporting on the treatment of uncomplicated appendicitis in children. We will also actively send out the final COS to relevant journal editors and funding bodies to promote uptake in future research“

VERSION 2 – REVIEW

REVIEWER	Francois Dos Santos Imperial College London United Kingdom
REVIEW RETURNED	13-Mar-2019

GENERAL COMMENTS	Very well structured and well written protocol. Perfect after the revisions made.
---

REVIEWER	Rebecca Fish University of Manchester, United Kingdom
REVIEW RETURNED	12-Mar-2019

GENERAL COMMENTS	In general, the authors have addressed the issues raised in the first round of reviews and added additional detail where requested. The detail on which parts of the project have already been completed already and the acknowledgement of persisting methodological uncertainty around involving children is a strength of the revised manuscript and demonstrates commendable transparency. The justification given in the discussion of why a global COS is needed beyond the UK COS is clear, however I would like to see some of this in the introduction to make it clear to the reader from
---

	the start (rather than the rather vague "to overcome any limitations of a COS focused on UK-specific surgical practice"). There are a few sentences that would benefit from re-phrasing for clarity and one or two grammatical errors in need of correcting. For example: line 68: A globally relevant set of core outcomes for paediatric uncomplicated appendicitis "assessed" in an international online Delhi study. (should probably be "derived" or "agreed" Line 112 "insured" should probably be "ensured" Finally, the members of the study management group alluded to in the methods (line 169) should be listed at the end of the manuscript. (unless this is the same as the COS development group, in which case standardise the nomenclature and should Hall be included?).
--	---

REVIEWER	Sarah Gorst University of Liverpool, UK
REVIEW RETURNED	29-Mar-2019

GENERAL COMMENTS	The authors have adequately responded to all comments. I believe this protocol is now acceptable for publication.
---

VERSION 2 – AUTHOR RESPONSE

Reviewer 2:

1: "The justification given in the discussion of why a global COS is needed beyond the UK COS is clear, however I would like to see some of this in the introduction to make it clear to the reader from the start (rather than the rather vague "to overcome any limitations of a COS focused on UK-specific surgical practice")."

Response 1: We agree that the rationale for an international COS deserves a clear explanation. That is why we added part of the points made in the discussion to the introduction to explain the aim of the study, as suggested.

Page 4 Line 101: "Recently a study protocol was published for developing such a COS in the United Kingdom[15]. Because of the differences between countries in treatment practices, resources and cultural aspects it was decided, in conjunction with the UK COS research group, that there is a need for an international COS, to be used in all trials assessing the treatment of acute uncomplicated appendicitis in children. The development of the current international protocol was performed in conjunction with the UK research group. Its principal investigator (NJ Hall) has been involved in its development and is part of the study management group."

2: "There are a few sentences that would benefit from re-phrasing for clarity and one or two grammatical errors in need of correcting.

For example: line 68: A globally relevant set of core outcomes for paediatric uncomplicated appendicitis "assessed" in an international online Delhi study. (should probably be "derived" or "agreed"

Line 112 "insured" should probably be "ensured"

Response 2: We critically re-read the phrasing and adjusted some errors, including your suggestions. The other changes were minor and can be found in the marked copy document.

Page 2 Line 63: "This protocol describes an international online Delhi study that should result in a globally relevant set of core outcomes for paediatric uncomplicated appendicitis. "

Page 4 Line 110: "Parent and patient representation was ensured through involvement of the Dutch patient and parent Foundation"

3: "Finally, the members of the study management group alluded to in the methods (line 169) should be listed at the end of the manuscript. (unless this is the same as the COS development group, in which case standardise the nomenclature and should Hall be included?)."

Response 3: The study management group is a smaller group within the steering committee (as mentioned on page 6 line 149. We added the members to the footnotes, as suggested.

Page 18 Line 553: "Study management group: Knaapen M, Hall NJ, Van der Lee JH, Butcher NJ, Offringa M, Bakx R, Gorter RR"

We hope that with these responses we have adequately addressed the concerns posed by the reviewers. Thank you in advance for considering this article for publication.